# Fast Detection of *Sclerotinia Sclerotiorum* on Oilseed Rape Leaves Using Low-Altitude Remote Sensing Technology

**DOI:** 10.3390/s18124464

**Published:** 2018-12-17

**Authors:** Feng Cao, Fei Liu, Han Guo, Wenwen Kong, Chu Zhang, Yong He

**Affiliations:** 1College of Biosystems Engineering and Food Science, Zhejiang University, 866 Yuhangtang Road, Hangzhou 310058, China; caofeng0702@163.com (F.C.); 3130100424@zju.edu.cn (H.G.); wwkong16@zafu.edu.cn (W.K.); chuzh@zju.edu.cn (C.Z.); yhe@zju.edu.cn (Y.H.); 2Key Laboratory of Spectroscopy Sensing, Ministry of Agriculture and Rural Affairs, Hangzhou 310058, China; 3School of Information Engineering, Zhejiang A & F University, Hangzhou 311300, China

**Keywords:** *Sclerotinia sclerotiorum*, oilseed rape, multispectral technology, thermal imaging technology, image fusion, machine learning

## Abstract

*Sclerotinia sclerotiorum*, one of the major diseases infecting oilseed rape leaves, has seriously affected crop yield and quality. In this study, an indoor unmanned aerial vehicle (UAV) low-altitude remote sensing simulation platform was built for disease detection. Thermal, multispectral and RGB images were acquired before and after being artificially inoculated with *Sclerotinia sclerotiorum* on oilseed rape leaves. New image registration and fusion methods based on scale-invariant feature transform (SIFT) were presented to construct a fused database using multi-model images. The changes of temperature distribution in different sections of infected areas were analyzed by processing thermal images, the maximum temperature difference (MTD) on a single leaf reached 1.7 degrees Celsius 24 h after infection. Four machine learning models were established using thermal images and fused images respectively, including support vector machine (SVM), random forest (RF), K-nearest neighbor (KNN) and naïve Bayes (NB). The results demonstrated that the classification accuracy was improved by 11.3% after image fusion, and the SVM model obtained a classification accuracy of 90.0% on the task of classifying disease severity. The overall results indicated the UAV low-altitude remote sensing simulation platform equipped with multi-sensors could be used to early detect *Sclerotinia sclerotiorum* on oilseed rape leaves.

## 1. Introduction

Plant diseases represent a significant threat to our food system, impacting on crop growth and quality. *Sclerotinia sclerotiorum* is a plant pathogenic fungus, and is one of the most serious diseases affecting oilseed rape. It can usually be found on tissues with high water content and close proximity to the soil [1]. Leaves, stems, flowers and pods are the sensitive parts that can be infected by *Sclerotinia sclerotiorum* [2,3]. According to statistics, *Sclerotinia* stem rot occurs in all oilseed rape production areas in China, and its incidence rate is generally 10% to 80%, meaning the annual output loss can reach up to 30% [4]. 

As we all know, leaves play an important role in plant growth, as they enable photosynthesis to occur. Photosynthesis is the process by which leaves absorb light and carbon dioxide to produce carbohydrates for plants to grow. Plant disease on leaves can cause serious consequences. Disease fungi takes their energy from the leaves on which they live. There is a negative relationship between disease injury and the photosynthetic efficiency of plants [5], which has bad effects on the growth of plants. Therefore, fast and accurate detection of diseases on plant leaves is of great practical value. 

Many non-destructive methods have been developed to detect diseases on plant leaves [6,7,8,9,10], and these methods have achieved good results. The disease process of plant leaves was shown to be associated with various changes in transpiration that depended on the stage of pathogenesis, and the transpiration rate was negatively correlated to leaf temperature [11,12]. As leaf temperature is related to the rate of evapotranspiration from the leaf surface, infrared sensing of the leaf temperature may be used to monitor the transpiration rate of plants. Infrared thermography is used to detect infrared energy emitted from objects, convert it to temperature, and display images of temperature distribution. This technology was previously used in military and industrial applications [13,14,15]. With the development of thermal imaging technology and the urgent need for non-contact temperature measurement, this technology has begun to be applied to agricultural detection. In recent years, this technology has been applied to crop growth monitoring [16,17] and disease detection [18,19,20,21,22]. Oerke et al. used digital infrared thermal imaging technology to detect and analyze the severity of apple leaf inoculation with apple black fungus at different times, as well as the effect of disease severity on the transpiration of apple leaves [20]. Lindenthal et al. studied the occurrence of downy mildew on cucumber leaves, and showed that transpiration of cucumber leaf tissue was correlated to leaf temperature in a negative linear manner (R = −0.762). Leaf areas developing symptoms of downy mildew in later stages of pathogen development exhibited a presymptomatic decrease in leaf temperature of up to 0.8 °C lower than non-infected leaves due to the abnormal stomatal opening [21]. Stoll et al. used infrared thermography to monitor irregularities in temperature at an early stage of pathogen development [22]. If thermal infrared technology is just starting out in agricultural non-destructive detection, then multispectral technology has been in use for decades. Multispectral imaging aims to acquire spatial and spectral information that are responsive to the surface and internal information of the object. Multispectral technology has a good effect on the detection of biochemical information of crop diseases, and it can directly reflect the internal structure and movement state of the molecule [23]. There have been some studies in agricultural, particularly for detection in agricultural product safety and quality [24,25], nutrient content [26,27] and plant disease [28,29,30]. 

At present, the disease severity is usually determined by artificial visual inspection and sampling in the field, these methods are time-consuming, labor-intensive, subjective, and not representative [31,32,33]. Using machine learning methods to classify disease severity levels can effectively solve these problems. The purposes of this study are as follows: (1) to investigate the potential of infrared thermography to distinguish the infected and non-infected areas of the attacked oilseed rape leaves by *Sclerotinia sclerotiorum*; (2) to evaluate the performance of four machine learning algorithms (SVM, RF, KNN and NB) in classification of different disease severity samples; (3) to explore the possibility of improving the classification results by image fusion based on multi-model images.

## 2. Materials and Methods

### 2.1. Plant Material

Oilseed rape seeds were sown in the experimental fields of Zhejiang University (Hangzhou, China). Uniformly sized seeds were transplanted into pots (diameter 18 cm) with a 2:1 mixture of organic soil and pine bark a month later at the 5-leaf stage. The seedlings were grown in a greenhouse at 18 ± 2 °C with the RH of 70 ± 10%. Plants were watered daily with water and fertilized weekly.

### 2.2. Sample Preparation

The *Sclerotinia sclerotiorum* were cultured on a potato dextrose agar. When the plants adjusted to the environment about a month after transplantation, mycelial pellets were placed on the leaves of oilseed rape and each leaf received two mycelial pellets symmetrically placed along the main vein. A total of 60 pots of oilseed rape were inoculated with *Sclerotinia sclerotiorum,* and 5 leaves were inoculated for per pot. 24 h later, mycelial pellets needed to be removed from the leaf surface when *Sclerotinia sclerotiorum* had infected the leaf tissues. Water was sprayed on the leaf surface twice a day to keep enough humidity around the disease spots. 

### 2.3. Data Acquisition and Preprocessing

The data acquisition system used in this study is presented in Figure 1. Thermal, multispectral and RGB images were acquired separately by corresponding imaging sensors. An indoor unmanned aerial vehicle (UAV) simulation platform was developed to mount these imaging sensors, which can move along approximately 1.5 m above the oilseed rape. Because the distance between the imaging sensors and the oilseed rapes was very short, if the three types of images were taken at the same time, it was unable to get the images from the same view due to the parallax. Therefore, the three sensors were mounted above the oilseed rape, and arranged in the same direction as the platform’s movements. The optical axes of the three sensors were adjusted to be perpendicular to the ground. Then the control program of the platform was set and images were taken when each sensor moved exactly above the corresponding oilseed rapes. This method could ensure that the images obtained from the three sensors from the same view. On the first day (Day 1), images of oilseed rape leaves were acquired before the inoculation. Every single image contains two pots of oilseed rape. The water on the leaf surface was wiped with absorbent paper before the data acquisition. On Day 2 to Day 6, approximately 24 h to 120 h after inoculation, the same method was used to acquire images.

Thermal imaging system consists of longwave infrared thermal camera (FLIR Tau2, Portland, OR, USA), acquisition card (IRSV-OTF, Hongpuweishi, Beijing, China), image transmission system (TS832, Haixun, Nanjing, China), remote control (AT9, RidioLink, Shenzhen, China), Monitor (FPV758, Fuweide, Zhangzhou, China), etc. FLIR Tau2 is a lightweight, uncooled, longwave infrared (LWIR) thermal imaging camera core [34]. It provides high-sensitivity (50 mK) infrared scanning at 640/8.7 Hz frame rates. This level of sensitivity allows for more accurate temperature measurements, and makes it ideal for telemetry and analytics. The temperature measurement module in FLIR Tau2 has been calibrated at the factory, and only a sample non-uniformity correction is required. IRSV-OTF acquisition card outputs 14-bit loss-less raw format digital images (gray value ranging from 0 to 16,383). The gray value of the acquired image is mainly concentrated in the range of 7000–7500 in the actual use scene.

In the procedure of thermal image processing, some numerical conversion is needed. The plateau histogram equalization algorithm seeks to maximize the dynamic range available for the content of the scene. A linear histogram equalization algorithm was used to translate the 14-bit thermal image into 8-bit space, which can protect the original temperature information. The translation was conducted according to the following equation:
(1)$$g\left(x,y\right)=\frac{255-0}{F_{max}-F_{min}}f\left(x,y\right)$$
where g(x,y) was the 8-bit image, f(x,y) was raw acquired image, $F_{max}$ was the maximum value of the raw acquired image, $F_{min}$ was the minimum value of the raw acquired image.

The temperature measurement was performed on the thermal image after the linear transformation using the temperature measurement formula. The obtained temperature distribution image t(x,y) was converted as follows:(2)t(x,y)=kf(x,y)−273.15
where k was the temperature measurement coefficient, and it was related to the thermal camera used.

A portable multispectral imaging system covering the spectral range of 600–875 nm was used to acquire multispectral image of oilseed rape leaves. A XIMEA xiQ multispectral camera (MQ022MG-CM, XIMEA, Munster, Germany) was used in this study. This multispectral imaging sensor is a new type of snapshot imaging sensor. It uses filter spectroscopy technology, and the filter is coated on the sensor pixels in the form of thin film. The principle is to set different bands of filters on the sensor as pixels, and to construct a unit with 5 × 5 = 25 pixels, that is, to coat 25 band-pass films with different central wavelengths on 25 pixels of a unit. The acquired multispectral image should be corrected from raw images to reflectance images by the following equation:(3)$$Ref_{a}=k_{i}\times DN_{a}+b_{i}$$
where $Ref_{a}$ was the reflectance measured by the a-th gradient target of the calibration plate, $DN_{a}$ was the a-th Digital Number (DN) of the calibration plate, $k_{i}$ was the gain of the i-th value, $b_{i}$ was the offset value of the i-th band which represents the response introduced by ambient radiation. Since the sensor responses can be approximated as linear, k equations were solved according to the least squares principle, and the final fitting coefficients $k_{i}$ and $b_{i}$ can be obtained according to the residual sum of squares (RSS). Due to the incompatibility between calibration document and the camera lens filter, only the data from Band13 to Band25 can be used. These bands are distributed between 730–875 nm, and they are located just near “red edge” where the reflectivity of leaves rises faster [35].

RGB images were captured by Canon EOS 650D in this study. It is an 18.0 megapixels digital single-lens reflex camera (DSLR) with a maximum resolution of 5184 × 3456. It is equipped with the DIGIC 5 processor, allows for a continuous shooting of 5.0 frames per second.

### 2.4. Image Registration and Fusion

The scale-invariant feature transform (SIFT) is a feature detection algorithm to detect and describe local features in images [36]. Feature extraction is a fundamental step for registration. The extracted features are invariant to scale and orientation, and are highly distinctive of the image. As the feature extraction methods can be used to identify the mutual information among the thermal, multispectral and RGB images, we can use this method to construct appropriate descriptors for finding matching pairs. Since there is no effective method for simultaneously matching the three types of images, the image registration was performed by finding matching pairs of thermal and multispectral images first, then thermal and RGB images. Hausdorff distance was used to find matching pairs [37,38,39]. It has obvious advantages compared with other distance measurements, for it only needs to calculate the maximum distance between two point sets rather than match the distance one by one. The Hausdorff distance which measures the similarity of the two point sets is defined as follows: (4)H(T, M)=max(h(T, M),h(M,T))
where T and M were two point sets of the thermal and multispectral images, h(T, M) was the directed Hausdorff distance from T to M defined as:(5)$$h\left(T,\;M\right)=\underset{t\in T}{\max}\;\underset{m\in M}{\min}d\left(t,m\right)$$
where d(t,m) was the Minkowski-form distance based on the $L_{P}$ norm, and defined as(6)$$d\left(t,m\right)={\left(\sum_{k}{\left(t_{k}-m_{k}\right)}^{P}\right)}^{1/P}$$

All the matched SIFT points were paired according to Algorithm 1 involving the computed Hausdorff distance H(T, M) which was used this time as the pairing threshold. The resulting doubly matched features then constitute the set R. 

**Algorithm 1** Matching AlgorithmGiven $T^{\prime }=T,\;M^{\prime }=M,\;R=\emptyset ,$

**for all**
$a_{i}\in A^{\prime }$
**do**
   **for all**
$b_{j}\in B^{\prime }$
**do**      **repeat**        **if**
$d\left(t_{i},m_{j}\right)=\underset{m\in M^{\prime }}{\min}d\left(t_{i},m\right)\wedge d\left(m_{j},t_{i}\right)=\underset{t\in T^{\prime }}{\min}d\left(m_{j},t\right)\wedge$        $d\left(t_{i},m_{j}\right)\leq H\left(T,M\right)\wedge d\left(m_{j},t_{i}\right)\leq H\left(T,M\right)$
        **then**
$\left(t_{i},m_{j}\right)\in M\wedge T^{\prime }=T\left\{t_{i}\right\}\wedge M^{\prime }=M\left\{m_{j}\right\}$        **end if**      **until**
$T^{\prime }\neq \emptyset \vee M^{\prime }\neq \emptyset$   **end for**
**end for**

**return**
R


To compensate for missing information from a single sensor, three types of images were stacked on the basis of thermal images. The resolution of thermal images is 640 × 512, the resolution of RGB images is 5184 × 3456, and the resolution of multispectral image is 407 × 207. Considering that the resolution of the three images is different, the registration process would scale the other two types of images to the size of the thermal image, where downsampling of the RGB image and interpolating of the multispectral image occurs. Finally, three types of images were merged into a 640 × 512 image with 17 bands, and the missing pixels with grayscale value of 0 were filled in the bands except for “thermal band”. It is a simple way of image fusion, which preserves the original information to the greatest extent possible for machine learning classification. The schematic diagram of image registration and fusion is shown in Figure 2, the process “a” represents registration of the infrared and multispectral images, the process “b” represents registration of the infrared and RGB images.

### 2.5. Machine Learning Models 

Support vector machine (SVM) is a widely used supervised pattern recognition method. It transforms original data into high-dimensional space, and constructs a hyperplane or multiple hyperplanes to maximize the distance between different categories of samples. It is very important for kernel functions to map original data to high-dimensional space in SVM. In order to get a better result, the optional penalty coefficient (C) and the kernel function parameter gamma (γ) were obtained by a grid-search procedure in the range of 2^−8^–2^8^ with the kernel function of RBF [40].

Random forest (RF) is an ensemble classifier, it creates a set of decision trees from randomly selected subset of training set. It then aggregates the votes from different decision trees to decide the final class of the test object [41]. The depth of the tree is very important. In this study, the depth of the tree was explored from 2 to 30.

K-nearest neighbor (KNN) is a non-parametric method used for classification and regression. KNN is a type of instance-based learning, or lazy learning, where the function is only approximated locally and all computation is deferred until classification. The KNN algorithm is among the simplest of all machine learning algorithms. It has two advantages: (1) Robust to noisy training data; (2) Effective if the training data is large [42,43]. In this study, the neighbors of the model was explored from 2 to 20.

The naïve Bayes (NB) classifier is an algorithm based on Bayesian theory in the classification algorithm set. Bayesian theory refers to calculating the probability of occurrence of another event based on the probability of an event that has occurred. The basic idea of NB is to calculate the probability with all possible values of the known class variable y, and choose the output probability to be the largest result, that is the predicted class label [44,45]. Mathematical expressions can be written as:
(7)$$\hat{y}=arg\;\underset{y}{\max}P\left(y\right)\prod_{i=1}^{n}P(x_{i}|y)$$
where P(y) denotes the class probability, $P(x_{i}|y)$ denotes conditional probability, $\hat{y}$ denotes the predicted class label. There had three different labels $y=\left\{y_{1},y_{2},y_{3}\right\}$ in this study, corresponding to three disease severity levels.

### 2.6. Schematic View of Research Methodology

There were two experiments conducted in this study. The schematic view of the two experiments is presented in Figure 3. Experiment 1 was concerned with the relationship between leaf temperature and pathogen infection, the maximum temperature difference (MTD) and temperature change curves were the two most important indicators. Experiment 2 was concerned with the performance of four machine learning algorithms (SVM, RF, KNN and NB) in the classification of different disease severity samples, and compared the classification of thermal dataset and fused dataset. More specific details of image acquisition and data analysis for detection of oilseed rape disease are presented in Figure 4.

## 3. Results

### 3.1. Relationship between Leaf Temperature and Pathogen Infection

In the first experiment, the interaction between *Sclerotinia sclerotiorum* infection and the oilseed rape leaves were detected thermographically on the second day after inoculation, before obvious visual symptoms occurred. Compared with non-infected leaves, the temperature distribution of the infected leaves showed obvious unevenness. The leaf temperature response differed in infected and non-infected areas as leaf tissues continue to be infected by *Sclerotinia sclerotiorum* during the following days. 

The maximum temperature difference (MTD) was studied by segmenting leaf regions from the thermal image manually. The MTD of leaves before and after infection shows a trend of rising first and then falling (Figure 5) in this experiment. The MTD of infected leaves is significantly higher than that of the same leaves before infection. In this study, the MTD was maximum on the second day after inoculation, reaching 1.7 °C. In the following days, the MTD of infected leaves was 0.4 °C higher than that before inoculation.

The temperature change curves through the disease spot before and after infection were plotted (Figure 6). They were drawn by selecting the temperature values of 40 pixels (white line in thermal pseudo-color images) through lesion area. As can be seen from the figure, the temperature curve before inoculation was very gentle, staying at around 16 °C. On the second day, the temperature difference was clearly visible between the lesion and other areas in the thermal image, while only slight yellow dots could be seen in the RGB image. The specific temperature difference was expressed as the temperature of the lesion area was significantly lower than other areas, and the temperature of the pre-infected area where lesion extends outward to the edge of leaf was higher than the other part of leaf. On the third day, the lesion area expanded, and necrotic area occurred in the center of the lesion area. Furthermore this part of the cell was completely necrotic. It was worth noting that the temperature of pre-infected area was 0.7 °C higher than the non-infected area. On following three days, the necrotic area continued to expand, while the temperature difference of each area was consistent with the previous two days.

On the whole, the relationship between leaf temperature and pathogen infection on oilseed rape leaves showed a regular phenomenon. As infection spreads, the disease spot gradually appeared in three areas: the necrotic area, lesion area and pre-infected area. The temperature of each area presents different characteristics, the areas from high to low levels were: necrotic area > lesion area > per-infected area. Necrotic area was the highest temperature zone in the leaf, which was easily detected. The temperature in the lesion area was the lowest in the whole leaf, the red arrows in Figure 6 point to the lesion area. The temperature in the pre-infected area was higher than the normal area. This indicates the potential of thermal imaging to generate specific signatures for plant-pathogen interactions, usable as a fingerprint for early disease detection on the oilseed rape leaf.

### 3.2. Thermal Image Preprocessing

The temperature difference of oilseed rape leaves in thermal image was easy to identify, and it showed a potential to detect disease, as well as to classify different disease severity levels using machine learning methods. We conducted another experiment, a total of 40 pots of oilseed rape were inoculated with *Sclerotinia sclerotiorum*, and three types of image were acquired. 32 × 32 pixels image in representative leaves was cut from each pot of oilseed rape as a data sample. Three levels were divided according to the days post infection (DPI). Level 0 stands for healthy samples (0 DPI), Level 1 and Levels 2 stand for mild infected samples (1 DPI) and severe infected samples (5 DPI) respectively. Three severity levels of sixteen oilseed rape leaves are shown in Figure 7. Min-max normalization was applied to preprocess the 32 × 32 pixels image. This normalization algorithm linearly transformed the raw data and scaled the data between the 0 and 1. Two fixed size images were cut from the acquired thermal image of each pot of oilseed rape in 0 DPI, 1 DPI and 5 DPI. After that, every 32 × 32 pixels image was converted to a 1024 size column vector, and a 240 × 1024 size vector was constructed as the thermal dataset after the above processes.

### 3.3. Classification Results on Thermal Dataset

Support vector machine (SVM), K-nearest neighbor (KNN), random forest (RF) and naïve Bayes (NB) were used to build classification models on the thermal dataset. In this study, accuracy was used to evaluate classification models. Informally, accuracy is the fraction of predictions the model got right. Formally, accuracy has the following definition: (8)$$Accuracy=\frac{Number\;of\;correct\;precictions}{Total\;number\;of\;predictions}$$

In order to get as much valid information as possible from the limited data, k-fold cross-validation was used to assess the predictive performance of the models. In this study, 5 times 5-fold cross validation was introduced, and the thermal dataset was divided into 5 groups. For each unique group, the group was taken as the training set, and the remaining groups were taken as the test set. It is necessary to randomly use different divisions repeated 5 times to reduce the difference introduced by the different samples. The results of these classification models are shown in Table 1. 

For the classification models, the classification accuracy is a key evaluation indicator. The SVM model obtained the best classification results in both the training set and the test set, with an accuracy of 99.38% and 78.33% respectively. However, the degree of overfitting was also the highest in the four models. The test accuracy of KNN, RF and NB models were almost the same, which was around 70%. The overfitting degree of the KNN and NB models was acceptable, much lower than the other two models. Differences could be observed from the performances of the same model between the different cross validation, the accuracy difference could be up to 15% (the training accuracy of 4 and 5 cross validation in KNN model). The SVM model had the longest training time of around 1290 s due to the use of grid search to find the optimal parameters. In comparison, NB was a probability-based generation model that took less time on the training process. There was a trade-off between the training time and accuracy in the classification of disease severity. We called it the “speed-accuracy trade-off”. If we wanted the classification process to be as fast as possible, the NB model was the best choice, as the training time was only 1/20,000 of SVM model. While the training time was not the focus of attention, the accuracy of SVM model was at least 8.3% higher than the other three models.

The classification accuracy of each model was different. In order to find out which of these samples were correctly classified and which were misclassified, the confusion matrix was used to evaluate the model. A confusion matrix is a table that is often used to describe the performance of a classification model on a set of dates for which the true values are known. The number of correct and incorrect predictions are summarized with count values and broken down by each class, and this is the key to the confusion matrix. The confusion matrices of the training set and test set in SVM, KNN, RF and NB models are shown in Table 2. The samples for three levels were randomly divided into a training set and a test set at a 2:1 ratio, with 160 samples in the training set and 80 samples in the test set. The training accuracy is 100% in this random division when using the SVM model, which meant that all the training set samples were correctly divided. However, the accuracy of the test set was not so high. A total of 12 samples in 80 samples were misclassified. For the 31 samples with the label “Level 0”, 29 were correctly divided, 2 were incorrectly divided, and the division accuracy of “Level 0” was 93.55%. This can be seen intuitively in Table 2. The “Level 0” samples had the highest accuracy, followed by “Level 1”, and finally “Level 2”. The healthy samples were relatively easy to find out from the three samples, but the overall classification was not satisfactory. The total accuracy is the weighted average of the accuracy of three different disease degree samples, and can reflect the overall performance of the model.

### 3.4. Classification Results on Fusion Dataset

The above analysis used the thermal image, which could reflect the temperature information of the leaves. However, some other important information (color, texture, etc.) could not be obtained from the thermal images. In this experiment, 3 types of images were acquired. The raw multispectral image had 25 bands, with the resolution of 407 × 217 for each band. 13 bands were finally retained after eliminating the band with abnormal noise. RGB image had 3 bands and thermal image had one band. After image fusion, we got a 17-band fused image, with the resolution of 640 × 512 for each band. Then, we used the similar method to get a 240 × 1024 × 17 vector as the fusion dataset. 

The visualization of confusion matrices using the thermal dataset and the fused dataset are shown in Figure 8 and Figure 9 respectively.

The classification accuracy of Level 1 and Level 2 was significantly improved when using the fused dataset. Although Level 2 was still the most difficult class to be correctly classified, it had about an 11% performance improvement over the previous dataset. Compared with KNN, RF and NB models, the performance of SVM models in classification accuracy was the best, with 94%, 89% and 87% in each class. The use of image registration and fusion improved the quality of the dataset, and achieved satisfactory classification results.

## 4. Discussion

The main goal of this study was to detect *Sclerotinia sclerotiorum* on oilseed rape leaves using low altitude remote sensing technology. Oilseed rape leaves were inoculated with *Sclerotinia sclerotiorum* artificially, three different types of images (thermal, multispectral and RGB image) were acquired before and after infection. The accuracy specification of the longwave infrared thermal camera (FLIR Tau2) was ±2 °C when the temperature of measured object was below 100 °C, which meant that comparing the absolute temperature of the measured object was worthless. In practical applications, the maximum temperature difference (MTD) was the most important index for analyzing plant leaf temperature differences. Although the infrared sensitivity specification reached 50 mK (0.05 °C) in FLIR Tau2, when the measured object itself was in thermal equilibrium or its ambient temperature difference was small, the sensor could not obtain the required temperature field information in most cases. This leads to a sharp rise in noise in the image, and the identifiability of the image was severely reduced, hence contributing to the temperature fluctuation in adjacent pixels [46]. This also showed that the MTD of non-infected leaves reached to 0.78 °C on average in this study. In comparison, multispectral technology had a good effect on the biochemical information detection of crop diseases, because it could respond to molecular structure and its internal state [47]. The portable multispectral imaging system was light and easy to be mounted on drones. However, the design of multispectral camera with high spatial resolution was confined by infrastructure limits in bandwidth transmission and manufacturing techniques [48]. The resolution of image acquired by multispectral camera (XIMEA xiQ) was only 407 × 217 for each band, which had a certain distance from the accurate detection of plant diseases in practical applications. Besides, there would be many noise points in the image when the light condition was not very good. These factors made XIMEA xiQ incapable of being used alone in disease detection. RGB image taken by digital camera had the highest image resolution, but the color difference between infected and non-infected tissues was tiny, and it was better to use the RGB image with other sensors. 

The thermal image was the main source of data in this study, supplemented by multispectral and RGB images. After timing the analysis of temperature on the same oilseed rape, we found that the MTD of thermal image can effectively show the difference in the early stage after inoculation, which was a good indicator for studying the disease on oilseed rape. With the continuous infection of *Sclerotinia sclerotiorum*, the disease spots could be divided into three areas, and the temperature of each area was different. Necrotic area was the highest temperature zone in the leaf, and related studies had shown the lack of transpiration reduced the leaf surface temperature and made this area the highest temperature zone in the leaf [21]. The temperature in lesion area was the lowest in the whole leaf. The possible reason was that due to the production of salicylic acid in this area, which led to a weakened metabolism, part of the leaf stomata closed and the respiration slowed down, thereby raising the leaf surface temperature in the pre-infected area [49]. From the time dimension, the oilseed rape leaves had different temperature at different stages of the disease. These factors made it possible to classify different disease leaves using machine learning methods. 

Machine learning methods played essential roles in classification tasks. Considering that it was difficult to get a large number of samples, we chose traditional machine learning methods for the classification tasks in this study. Support vector machine (SVM), K-nearest neighbor (KNN), random forest (RF) and naïve Bayes (NB) are four representative machine learning classification models. γ and C are two key parameters for the radial basis function (RBF) kernel SVM. Intuitively, the γ parameter defines how far the influence of a single training sample reaches, with low values meaning “far” and high values meaning “close”. The C parameter trades off correct classification of training samples against maximization of the decision function’s margin, a lower C will encourage a larger margin, therefore a simple decision function at the cost of training accuracy [50,51]. Grid search were used to tune the parameters (γ and C) of the SVM model, in order to select a set of parameters that the model performed best. The main idea of the KNN model is to determine its own category based on neighbor categories that are close in distance, so the parameter that usually needs to be tuned is the number of neighbors. RF is a bagging method in ensemble learning, the maximum number of iterations of the weak learning is the most important hyperparameter in the parameter tuning process. Unlike the above three models, NB model is a generative algorithm with the advantage of a short operation time and clear mathematical principle. Although the parameters were tuned to the four models, satisfactory results were never obtained, mainly showing a higher degree of overfitting and lower classification accuracy. 

These performances might be caused by the “bad dataset”. In this study, the thermal dataset was so-called “bad dataset”. It had the following two characteristics: high degree of overfitting and low classification accuracy, no matter which classification algorithm was used. This dataset was obviously not ideal. One solution was to rebuild a new dataset. This was also the main reason for using image registration and fusion to generate the fused dataset. The results showed that the fused dataset increased the average classification accuracy on test set, especially in the classification on mild and severe infected samples. The temperature difference between the two samples was relatively small, and the thermal images were not sufficient to distinguish them well. However, the difference between color and texture of the disease spots was relatively large, and these were reflected in multispectral and RGB images. Therefore, adding multispectral and RGB images as supplementary information in the dataset could significantly improve the classification accuracy of the samples.

## 5. Conclusion

In this study, an indoor UAV low-altitude remote sensing simulation platform equipped with multi-sensors was used to detect *Sclerotinia sclerotiorum* on oilseed rape leaves. The performances of four machine learning models (SVM, RF, KNN and NB) were evaluated using thermal images and fused images respectively, and the satisfactory detection results indicated that the fusion of multi-sensors was important and would help to bring remote sensing to real-world application. In future research, more samples and field experiments will be studied under different types of remote sensing images to detect *Sclerotinia sclerotiorum* on oilseed rape, as well as other crops.

## Figures and Tables

**Figure 1 sensors-18-04464-f001:**
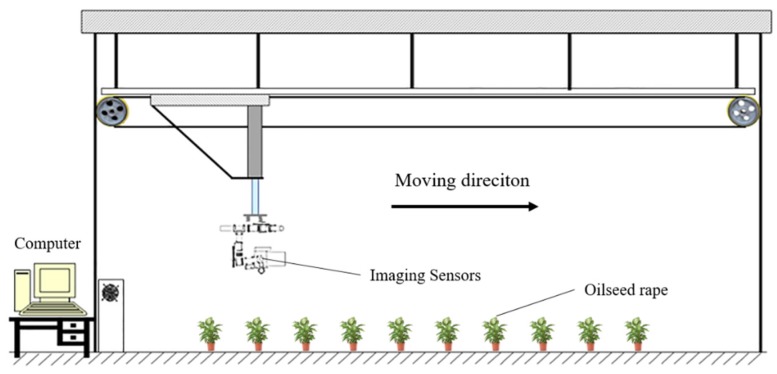
Schematic diagram of data acquisition system for disease detection.

**Figure 2 sensors-18-04464-f002:**
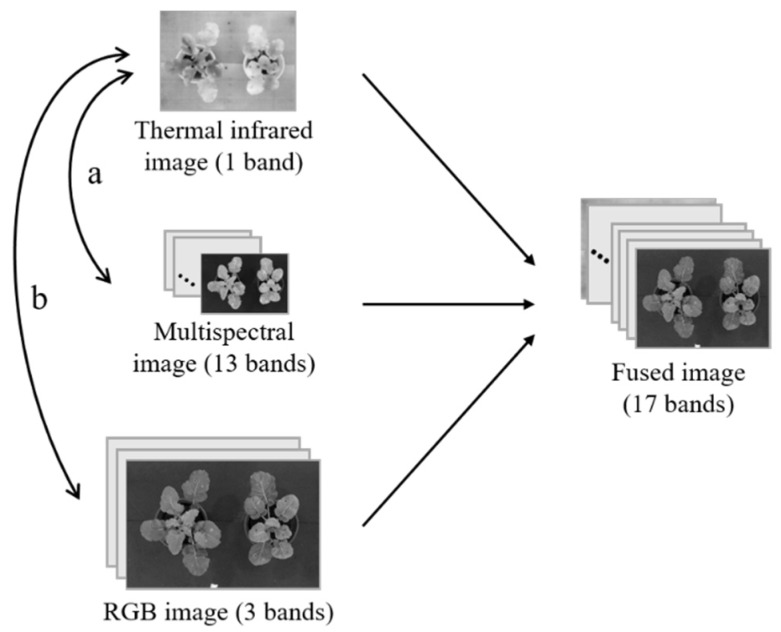
Schematic diagram of image registration and fusion.

**Figure 3 sensors-18-04464-f003:**
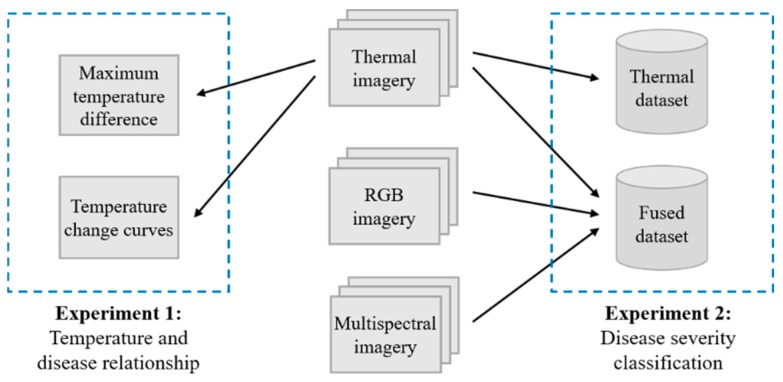
Schematic view of the two experiments. Experiment 1 was concerned with temperature and disease relationship, while experiment 2 was concerned with disease severity classification.

**Figure 4 sensors-18-04464-f004:**
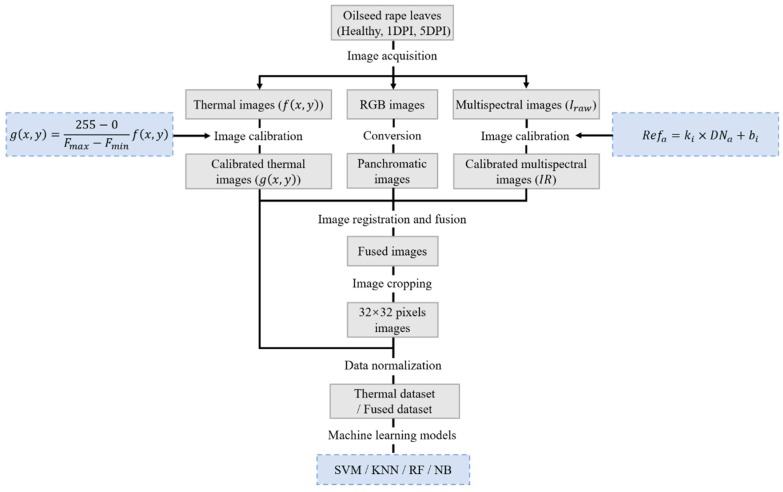
Flowchart of image acquisition and data analysis for detection of oilseed rape disease. Healthy, one day post infection (1DPI) and five days post infection (5DPI) oilseed rape leaves were the objects to be classified in this experiment.

**Figure 5 sensors-18-04464-f005:**
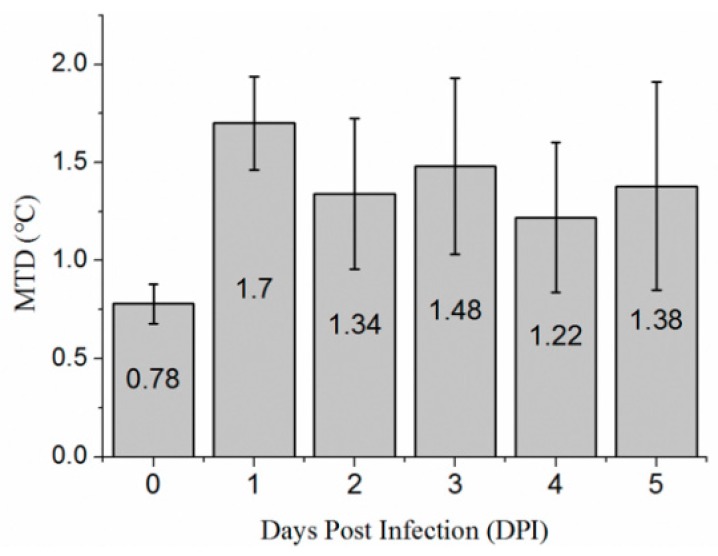
The maximum temperature difference (MTD) within leaves before and after infection. 10 leaves inoculated with *Sclerotinia sclerotiorum* at day zero were counted.

**Figure 6 sensors-18-04464-f006:**
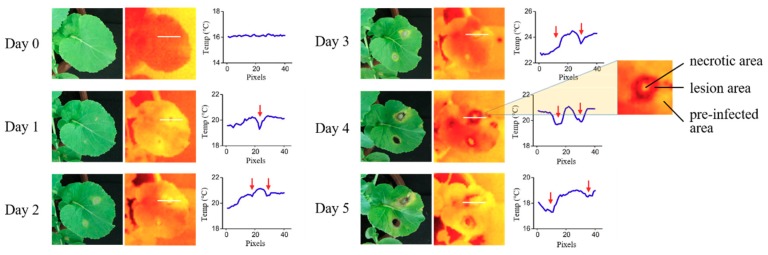
The temperature change curves through the lesion area before and after infection. Three areas (necrotic, lesion and pre-infected areas) gradually appeared in the disease spot.

**Figure 7 sensors-18-04464-f007:**
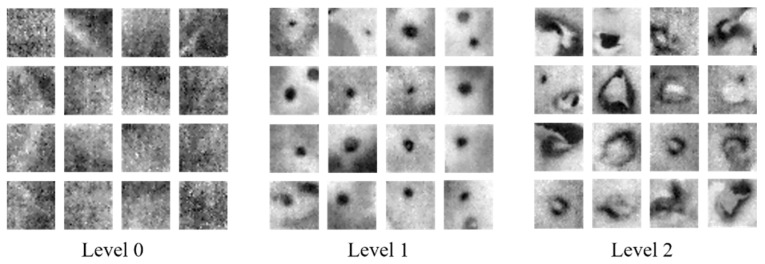
Three disease severity levels: Level 0 for healthy samples, Level 1 for mild infected samples and Level 2 for severe infection samples.

**Figure 8 sensors-18-04464-f008:**
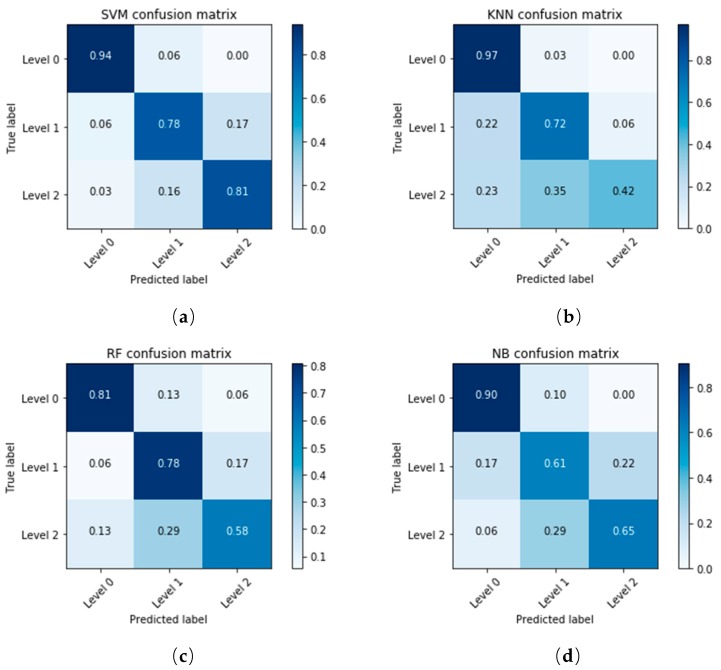
The confusion matrices of SVM, KNN, RF and NB models using the thermal dataset.

**Figure 9 sensors-18-04464-f009:**
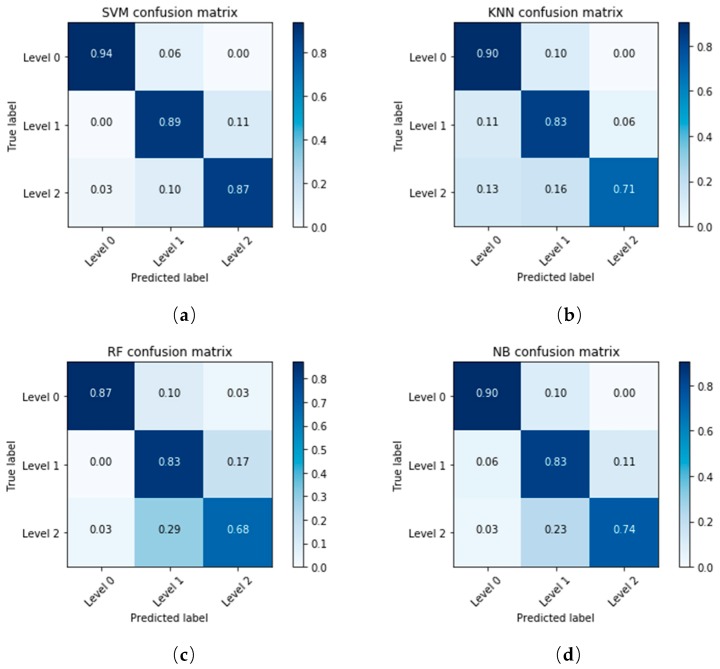
The confusion matrices of SVM, KNN, RF and NB models using the fused dataset.

**Table 1 sensors-18-04464-t001:** The results of the support vector machine (SVM), K-nearest neighbor (KNN), random forest (RF) and naïve Bayes (NB) models on the thermal dataset, with 5 times 5-fold cross validation.

Models	Time (s)	Training Accuracy (%)						Test Accuracy (%)					
		1	2	3	4	5	Avg	1	2	3	4	5	Avg
SVM	1290.83	98.96	100	99.48	98.44	100	99.38	75.00	83.33	72.92	81.25	79.17	78.33
KNN	1.64	78.13	73.43	78.13	83.33	68.75	76.35	66.67	72.92	75.00	75.00	64.58	70.83
RF	17.01	90.10	92.71	99.48	86.98	82.81	90.42	64.58	75.00	70.83	60.47	79.17	70.01
NB	0.07	82.29	75.52	79.17	75.52	80.72	78.64	68.75	72.92	70.83	68.75	70.83	70.41

**Table 2 sensors-18-04464-t002:** The confusion matrices of the training set and test set in SVM, KNN, RF and NB models.

Models		Training Set				Test Set			
		1	2	3	Accuracy (%)	1	2	3	Accuracy (%)
SVM	1	49	0	0	100	29	2	0	93.55
	2	0	62	0	100	1	14	3	77.78
	3	0	0	49	100	1	5	25	80.65
	Total				100				85.00
KNN	1	49	0	0	100	30	1	0	96.77
	2	9	51	2	82.26	4	13	1	72.22
	3	12	6	31	63.27	7	11	13	41.94
	Total				81.88				70.00
RF	1	46	1	2	93.88	25	4	2	80.65
	2	4	55	3	88.71	1	14	3	77.78
	3	2	10	37	75.51	4	9	18	58.06
	Total				86.25				71.25
NB	1	46	3	0	93.88	28	3	0	90.32
	2	7	43	12	69.35	3	11	4	61.11
	3	7	11	31	63.27	2	9	20	64.52
	Total				75.00				73.75

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
