# Peer review of "Fast Detection of Sclerotinia Sclerotiorum on Oilseed Rape Leaves Using Low-Altitude Remote Sensing Technology"

_sensors, 2018, doi:10.3390/s18124464_

Reviewer 1 Report

This manuscript presents a comparative study of different classification methods for the detection of disease infection on oilseed rape based on thermal, multispectral or rbg images.

The protocol of acquisition is first presented with an indoor aerial vehicle transporting imaging sensors. In this section, some information are missing on what has been made so that the 3 sensors register the same views of the plants. The multispectral camera seems to have up to 25 bands but it is not clear which one are used and to which spectral band they correspond. Some comments on that are made on section 'results' but comes a bit late in the text.

The methods section then describes first the alignement method based on SIFT feature detection. Some information are missing on how the different image are registered (simple translation?) and fused together. In particular they are defined at different resolution. It is not clear which final resolution is chosen and how the different images are transform into this final resolution.

In the result section, a first part is dedicated to show the effect of infection on leaf temperature based on analysis of thermal image. It is not clear however if thermal images are sufficient or not to detect level of infection.

The second part of this section shows results of learning and automatic classification of the different considered classification methods (SVM, KNN, Random Forest and Naive Bayes).  Using accuracy (whose definition is missing) and confusion matrix (same) a comparison between the different methods is proposed. From these results, SVM seems to be the most performant method.

The discussion section discuss the interest to complement thermal imagery with multispectral one and the difference between the classifier in term of results. In this part, the authors mention that performance can be bias by a "bad dataset". It is not clear for me why this dataset has some flaws. This should be discuss more precisely.

Author Response

We uploaded the response file in the attachment.

Reviewer 2 Report

In my opinion, the overall paper is not well written to read and understand. The results provided in this paper are interesting, but the paper is technically incorrect without proper content (lots of content and equations mismatching) and lack of proper references. Algorithmic steps should be presented for the simulated datasets. In my opinion, the paper can only publish after extensively addressing following comments,

1.      The structure of the manuscript is confusing. I suggest the authors to add a schematic view of the used methodology in order to clarify the content.

2.      I suggest the authors to explain more about the image pre-processing. Did you used any data correction? What are the pre-processing that you have applied to the images?

3.      All mathematical terms should be properly explain on it’s first use, e.g.  eqn. (3) to eqn. (5)

4.      The eqn. (6) at line 192, I think that there is a problem with the notation. It seems that the value of parameter y was chosen arbitrarily, Please elaborate more on your decision to use y.

5.      How is naïve Bayes (NB) selected for classification-based relative sclerotinia sclerotiorum estimation? 

6.      How you chose the values for “K-nearest neighbor (KNN)” in the proposed model?

7.      On what basis you minimized the mean absolute percentage error of prediction for disease estimation accuracy?

8.      “There was a trade-off between the training time and accuracy in the classification of disease severity”. How much is the accuracy trade-off, any value/percentage?  

9.      How you define the k-fold cross validation to simplify the model framework?

10.  What threshold values of overall accuracy you considered in data fusion statistic?

11.   What is the observation characteristic of combining all three sets of data? Please define properly in the text.

12.  Define the formulation of model used in Figure 5.

13.   Present an explicit and clear algorithmic steps used in this study data simulation.

14.  Use the high resolution image for Figure-5 and Figure-6.

15.  There is a need of couple of more proper reference to support this study.

Also, the paper should be proofread for sentences flow, English grammar correction, and spelling mistakes.

Author Response

(The authors gave the same response as above.)

Reviewer 3 Report

The article was focused on the rapid detection of Sclerotinia Sclerotiorum on oilseed rape leaves using the low-altitude remote sensing technology. The subject of the research was rape seed oil. The presented article was therefore focused on evaluation the potential of infrared thermography to distinguish the infected and non-infected areas of the attacked oilseed rape leaves by Sclerotinia sclerotiorum; further on to evaluate the performance of four machine learning algorithms (support vector machine, random forest, K-nearest neighbor and naïve Bayes) in classification of different disease severity samples; and in the third row on to explore the possibility of improving the classification results by image fusion based on multi-model images. Thermal, multispectral and RGB images were acquired separately by corresponding imaging sensors. A simulation platform for unmanned aerial vehicles has been developed to fit these imaging sensors. The authors described all three systems (thermal imaging system, portable multispectral imaging system and RGB image) without a detailed description of each component. Since there is no effective method for simultaneously matching the three types of images, the image registration authors have done by finding matching pairs of thermal and multispectral images first, then thermal and RGB images. The authors also described the machine learning models. The results confirmed this the potential of thermal imaging to generate specific signatures for plant-pathogen interactions, usable as a fingerprint for early disease detection on oilseed rape leaf. The results further showed that one type of image capture was not sufficient, and that an important parameter was the maximum temperature difference (the thermal image at the initial stage of the difference after inoculation can effectively demonstrate the difference). With the continuous infection of Sclerotinia sclerotiorum, the disease spots could be divided into three areas (necrotic, lesion and pre-infected areas), and the temperature of each area was different. The results shown that the fused dataset increase the average classification accuracy on test, especially in the classification on mild and severe infected samples.

 Strengths side

- detailed methods used

- the results are appropriately evaluated

 Weaknesses side

- a simple description of the sensors used - I propose to describe in detail the components of the device

 Other remarks:

- the abbreviations used in the tables and figures must be described in the legend

- I propose to add more detailed sensor parameters.

- Key words - Please delete the irrigation. The article does not deal with irrigation.

Author Response

We uploaded the response file in the attachment.

Round  2

Reviewer 1 Report

line 232 : ditails -> details

Reviewer 2 Report

Thanks for incorporating all previous comments and clarifying all doubts. I accept this manuscript with the MDPI publishing standards.